# An Inertial Sensor-Based Gait Analysis Pipeline for the Assessment of Real-World Stair Ambulation Parameters

**DOI:** 10.3390/s21196559

**Published:** 2021-09-30

**Authors:** Nils Roth, Arne Küderle, Dominik Prossel, Heiko Gassner, Bjoern M. Eskofier, Felix Kluge

**Affiliations:** 1Machine Learning and Data Analytics Lab (MaD Lab), Department of Artificial Intelligence in Biomedical Engineering (AIBE), Friedrich-Alexander-Universität Erlangen-Nürnberg (FAU), D-91052 Erlangen, Germany; arne.kuederle@fau.de (A.K.); dominik.prossel@fau.de (D.P.); bjoern.eskofier@fau.de (B.M.E.); felix.kluge@fau.de (F.K.); 2Department of Molecular Neurology, University Hospital Erlangen, D-91054 Erlangen, Germany; heiko.gassner@uk-erlangen.de; 3Fraunhofer Institute for Integrated Circuits IIS, D-91058 Erlangen, Germany

**Keywords:** HMM, IMU, segmentation, ascending, descending, classification, trajectory, ETKF, ZUPT, free-living

## Abstract

Climbing stairs is a fundamental part of daily life, adding additional demands on the postural control system compared to level walking. Although real-world gait analysis studies likely contain stair ambulation sequences, algorithms dedicated to the analysis of such activities are still missing. Therefore, we propose a new gait analysis pipeline for foot-worn inertial sensors, which can segment, parametrize, and classify strides from continuous gait sequences that include level walking, stair ascending, and stair descending. For segmentation, an existing approach based on the hidden Markov model and a feature-based gait event detection were extended, reaching an average segmentation F1 score of 98.5% and gait event timing errors below ±10ms for all conditions. Stride types were classified with an accuracy of 98.2% using spatial features derived from a Kalman filter-based trajectory reconstruction. The evaluation was performed on a dataset of 20 healthy participants walking on three different staircases at different speeds. The entire pipeline was additionally validated end-to-end on an independent dataset of 13 Parkinson’s disease patients. The presented work aims to extend real-world gait analysis by including stair ambulation parameters in order to gain new insights into mobility impairments that can be linked to clinically relevant conditions such as a patient’s fall risk and disease state or progression.

## 1. Introduction

The ability to climb stairs is a fundamental part of independent daily living and community participation [1]. In contrast to level walking, stair ambulation can be a more challenging task as it puts increased emphasis on lower limb muscle strength [2], requires a higher range of motion at lower limb joints [3], and increases demand on the postural control system [4]. Furthermore, psychological factors such as the fear of falling can additionally affect stair ambulation abilities [5]. Reasons for poor stair negotiation abilities can include functional disabilities such as dyspnea, but also neurological disorders which affect the motor system [2,6]. Conway et al. [2] found that PD patients showed a significantly slower gait speed while ascending and descending an instrumented staircase, which is likely to be related to an increased risk of falling during stair negotiation. At the same time, stairway falls may result in disproportionately worse injuries or even death compared to level-walking falls [4]. Hence, for patients who are already at a higher risk of falling due to PD, multiple sclerosis, or stroke [7], an objective assessment of stair climbing performance could be of great interest, providing support in disease management.

Several technologies and methods have been proposed in the past to analyse and quantify human gait. Measurement systems range from stationary vision-based systems, pressure carpets, or force plates to body-worn mobile sensors such as inertial measurement units (IMUs) or pressure insoles [8]. IMU-based, mobile gait analysis systems in particular are gaining an increasing amount of interest for use in long-term and out-of-lab assessments due to the advancements in sensor technology and signal-processing algorithms [9,10]. During such real-world studies, wearable IMUs can extend the snapshot assessments of laboratory measurements with continuous and more natural insights into a person’s mobility and health status, or even details about disease progression or fluctuations [11].

However, even though objective stair-climbing performance is evidently a clinically meaningful parameter and a fundamental part of daily life activities, recent continuous real-world gait analysis studies [12,13] did not include stair ambulation parameters into their analysis. This emphasises the need for adapted algorithms and gait analysis pipelines that are able to simultaneously segment, parametrize, and classify different stride types such as level walking, stair ascent, and stair descent using continuous inertial sensor data. New solutions should enable an individual analysis of these daily life activities in terms of macro gait parameters such as the number of strides or bout length, but also micro gait parameters, including gait velocity or swing and stance phase variations.

To extract objective stride parameters from continuous IMU data streams, the first part of most foot-worn sensor-based gait analysis systems is the segmentation of individual strides. Stride segmentation approaches range from peak detection [14] and template matching [15] to probabilistic data-driven approaches such as the hidden Markov models (HMMs) [16,17,18] or the deep learning-based models [19,20]. However, although some of the mentioned studies included stair-walking sequences in their datasets, stair-ascending and descending strides were not included within the evaluation of segmentation performance or were rejected as non-walking sequences by the algorithms on purpose. While those methods might be valid for laboratory assessments where only level walking strides occur, improved and more flexible segmentation methods are required for continuous real-world IMU data. During daily life recordings, gait sequences are likely to contain a mixture of different stride types, including stair ambulation sequences, which need to be segmented along with level-walking strides for a subsequent parametrization.

To enable a more detailed insight into gait performance, temporal parameters such as swing and support times can be extracted from the identified strides, which contain clinically relevant information about balance or asymmetry. Respective gait events such as the initial contact (IC) and the terminal contact (TC) are of particular interest for the calculation of these parameters [21]. Several studies have presented different signal features from foot-worn IMU data that detect those events [22,23,24]. Still, those approaches were often only evaluated in controlled level walking, a limited number of stair strides and stair configurations, or only in a single activity, either ascending or descending. Therefore, new and adapted algorithms for gait event detection are necessary, which will enable an accurate gait event detection independent of gait activities such as level walking, stair ascending, or stair descending. Furthermore, new methods must prove their reliability across varying environmental conditions such as changing walking speeds or different stair geometries, as expected in real-world studies.

In order to allow an individual analysis of gait performance for level walking as well as for stair ascent and descent, the respective stride types need to be classified. While human activity classification is a wide research field on its own, these approaches are often based on sliding windows with fixed length and focus primarily on macro parameters such as the time spent in individual classes [25]. In the literature, abstract signal features such as the wavelet transform [26] or the phase variable approach [27] as well as spatial trajectory features [28] were proposed for individual stride-typeclassification. Song et al. [28] successfully classified level walking, ramp walking, and stair walking during an outdoor walking course based on spatial stride features. They showed good classification accuracy for steady-state stair strides but reported limitations, especially for transitions between activities when single-step height strides occurred. While using spatial features to distinguish stair strides from level-walking strides might be an obvious approach, improved and accurate trajectory-tracking methods are required to tackle the limitations during transitions and to enable a robust classification even for single-step stair strides.

Although a wide range of research articles and studies in the field of mobile IMU-based gait assessment exists, respective analysis pipelines are often restricted to level-walking gait and do not enable an individual analysis of other gait-related activities. Especially for continuous real-world gait studies where stair ascending and descending sequences are likely to be expected during specific walking bouts, adapted algorithms and novel combined analysis pipelines assessing digital mobility outcomes (DMOs) are required. New DMOs in terms of detailed stair ambulation parameters such as stance or swing time variability during stair ascent or descent are still missing from real-world gait analysis studies.

To address these limitations, we present the following contributions:We present a new algorithmic pipeline that can extract stride-level temporal parameters from continuous IMU data, including level-walking and, specifically, stair-ambulation sequences.To enable a robust detection of individual strides within mixed gait-related activities, we extended an HMM-based segmentation approach by adding additional stride models to detect stair strides along with level-walking strides in continuous IMU data.The stride segmentation was combined with an adapted gait event detection algorithm to reliably estimate terminal and initial contact in the absence of a heel strike, which is often missing during stair walking.Stride-type classification was achieved through spatial features derived from a Kalman filter-based walking bout trajectory reconstruction and a subsequent walking bout assembly to ensure the high precision of the classification.Each individual pipeline step was evaluated on a new recorded dataset of 20 healthy participants walking on various stair configurations and at different speeds to enable a wide variety of strides, as expected during real-world studies. In total, around 16,000 stair strides and 13,000 level-walking strides, including pressure insole reference and video annotations, were available for evaluation.The entire pipeline was additionally validated end-to-end on the 20 healthy participants and additionally on an independent dataset of 13 PD patients.

The presented work aims to complement existing sensor-based gait analysis systems, to address clinically motivated questions such as the assessment of patients’ fall risk [29] or disease state [30] for upcoming studies involving continuous real-world gait analysis.

## 2. Materials and Methods

### 2.1. Dataset

For the evaluation and validation of the proposed analysis pipeline, two individual datasets were recorded. All participants gave written informed consent prior to the recording. The study was approved by the local ethics committee (Friedrich-Alexander-University Erlangen-Nuremberg, Germany) Re-No. 106_13B.

#### 2.1.1. Evaluation Dataset

The dataset for the development and evaluation of the presented pipeline was recorded in a real-world outdoor environment with 20 healthy participants (Table 1). The study took place in a public area next to the Erlangen University Hospital where different stair configurations were present. To enable a variety of stair strides as found in real-world applications, three different staircases with diverse geometries (inclinations of 7.6°, 22.5°, and 33.4°) and a ramp were included in the dataset (Figure 1).

To further increase the heterogeneity of the dataset, the participants were asked to walk each staircase at self-selected normal, slow, and fast speeds. To enable a natural transition between stair and level walking, the participants were asked to take approximately three strides of level walking before and after the stairs. An additional trial was recorded where the participants started with continuous level walking, passed the staircase on their way as they would in their everyday life, and continued walking to some previously defined landmarks. Afterwards, they returned to the starting point on the same way, passing the stairs again in the opposite direction. In total, each participant completed 21 tasks (three staircases × up and down × three speeds + three staircases during continuous walking in preferred speed, including up and down), which involve around 20 min of combined walking activities. Throughout the manuscript, this dataset is referred to as the *evaluation dataset*.

#### 2.1.2. Validation Dataset

The second dataset was recorded to validate the proposed analysis pipeline and to test its performance on elderly patients with impaired gait. Therefore, 13 PD patients (Table 1) were included in the study as the second independent dataset. The PD patients, who had been diagnosed according to the guidelines of the German Society for Neurology (Hoehn and Yahr stage I-III), were required to be able to walk 4×10 m without support and the need for walking aids during their daily life. The PD patients were asked to walk one transition of floors (up and down) on a staircase inside the University hospital (Figure 1), starting and ending with two to three level-walking strides. Due to the presence of gait impairments, PD patients performed the stair-walking task only at their preferred speed. The usage of the handrail was explicitly allowed to ensure a safe transition between floors. Throughout the manuscript, this dataset is referred to as the *validation dataset* and was not included in the training or optimization process.

### 2.2. Sensor System and Measurement Setup

The IMU sensor units (Portabiles GmbH, Erlangen, Germany) featured a 3D accelerometer (range ± 16 g), a 3D gyroscope (range ±2000 degree per second (dps)) (BMI160, Bosch Sensortec GmbH, Reutlingen, Germany) with Bluetooth Low Energy connectivity, and an internal flash memory for stand-alone operation. Data were recorded with a sampling frequency of 204.8 Hz. IMU sensors were calibrated prior to the recordings using the method described by Ferraris et al. [31]. To enable an additional reference for foot ground contact, a custom force-sensitive resistor (FSR)-based insole and signal-conditioning circuit (non-inverting amplifier) were developed, which were connected via an extension ribbon cable to an available three-channel, 16-bit analog-to-digital converter (ADC) of the sensor units. Two sensor units were attached to the left and right shoe instep position using 3D-printed clips, while the insole conditioning circuit was fixed via a second clip on the lateral side of each shoe (see Figure 2). As an additional reference, all gait sequences were filmed during the study with a hand-held video camera by the study coordinator, with the permission of each participant.

### 2.3. Coordinate Transformation

Due to the attachment of the sensor to the shoe’s instep position, a rough alignment between sensor and shoe was given. However, to ensure a constant alignment between the sensor and the body coordinate system, across participants and datasets or even between studies, a coordinate transformation of the recorded IMU data was performed, as illustrated in Figure 3. To enable a comparable sensor orientation, the sensor coordinate system was aligned to the gravity vector $\vec{e_{g}}$ direction for each gait sequence. Therefore, the orientation of the sensor with respect to gravity was estimated by averaging all acceleration vectors during static periods $\vec{a_{s}}$. Static sequences between walking bouts (where participants were standing before the first or after the last stride) were defined as windows of minimum 1 s where the angular velocity of the foot was below 2.5 dps. The rotation vector $\vec{r}$ for the sensor-to-gravity alignment was defined by the rotation angle α around the vector $\vec{\theta }$:(1)$$\begin{array}{c} \vec{a_{s}}=\left(\begin{array}{c} a_{x}\\ a_{y}\\ a_{z} \end{array}\right),\\ \;\vec{e_{g}}=\left(\begin{array}{c} 0\\ 0\\ 1 \end{array}\right) \end{array}$$(2)$$\begin{array}{c} \vec{\theta }=\vec{a_{s}}\times \vec{e_{g}},\;\alpha =\arccos\left(\frac{\vec{a_{s}}\;\cdot \;\vec{e_{g}}}{∥\vec{a_{s}}∥*∥\vec{e_{g}}∥}\right) \end{array}$$(3)$$\begin{array}{c} \vec{r}=\vec{\theta }\alpha \end{array}$$

Furthermore, in order to handle the left and right foot sensor data with the same pipeline, sensor axes were transformed into a shared body coordinate system. Therefore, sensor axes will be referred to as **ml** (medial to lateral), **pa** (posterior to anterior), and **si** (superior to inferior), which correspond to the body coordinate system after gravity alignment (see Figure 3).

### 2.4. Reference Labels

#### 2.4.1. Stride Borders

Although stair ascending and descending strides follow similar repetitive swing- and stance phase patterns compared to level walking, the angular velocity within the mediolateral axis ($gyr_{ml}$) follows inherently different and stride type-specific patterns (Figure 4), which corresponds to the foot’s flexion during the gait cycle. While ascending, participants tended to show a forefoot walking behaviour, sometimes with no heel contact at all. Likewise, while descending, the foot rolling behaviour was often missing, resulting in the initial contact with toes first or the complete flat foot [32]. Nevertheless, all three stride types showed a prominent negative peak in the $gyr_{ml}$ axis, which refers to the foot lift or terminal contact (TC) [23]. This negative peak was used for manual stride border annotation (along with the video recordings of the feet), as suggested by previous literature [15,17,18]. To enable a consistent annotation process, the following criteria were applied: to ensure a stable minimum for the TC label, the negative peak had to have a minimum dip of −20 dps, and a stride was only considered valid if the swing maximum was at least 50 dps.

#### 2.4.2. Stride Class Labels

For each annotated stride, a stride-type label was added—*level walking* (which also included the ramp strides), *stair ascending*, or *stair descending* (Figure 4). Based on the video data, additional reference elevation levels were added that corresponded to the respective stair step height, which was known due to the specific stair configurations (Figure 1). During steady-state stair climbing, this value mostly corresponded to twice the stair step height. Therefore, these strides will be referred to as *double stair step strides*. During transitions between stair and level walking or on stair landings, single stair step height strides occurred, which will be referred to as *single stair step strides*. For level walking, the height difference was annotated to be zero, and for the ramp, a constant stride elevation of the ramp height divided by the number of ramp strides was assumed.

#### 2.4.3. Pressure Insole Reference

Using FSR sensors is a common approach to evaluate gait events and respective temporal gait parameters such as the swing and the stance time without the need for stationary equipment [33,34,35]. Gait events, including IC and TC, can be extracted from pressure sensor data by using thresholding methods [35,36]. Therefore, FSR sensors are commonly integrated into insoles or directly attached to the feet. Although the number and exact placement of FSRs can vary between studies, the most common positions include the heel and forefoot (toes or ball of the foot) to reliably detect IC and TC [21].

In our study, an insole with a custom pressure sensor was developed. Each insole was equipped with three FSR sensors ( 40 mm× 40 mm, RP-S40-ST). The sensors were placed at the heel, the first metatarsal head (MTH), and at the location of the big toe. As FSR sensors show an exponential change in resistance when linear force is applied, a non-inverting amplifier conditioning circuit was used:(4)$$V_{Out}=V_{Ref}\;\cdot \;\left(1+\left(\frac{R_{Ref}}{R_{FSR}}\right)\right)$$

$V_{Ref}$ was fixed at 0.1 V and $R_{Ref}$ was selected individually for each FSR during a calibration procedure prior to the study. Therefore, each FSR was loaded from 0.5 kg to 20 kg in 0.5 kg steps, using a load cell as reference. Although the conditioning circuit improved the linearity of the FSR sensor output, an individual calibration function was added to the measured voltage signal, which corresponded to a 6th order polynomial fit across the 0– 20 kg calibration measurement.

#### 2.4.4. Reference Gait Events

To convert the measured pressure data into reference gait events, the sum of all three pressure sensors at a given sample was considered to enable a robust event detection invariant of individual foot rolling behaviour (Figure 5). Therefore, TC and IC could be detected reliably independent of the heel or toes touching or leaving the ground first.
(5)$$\mathrm{Total}\;\mathrm{weight}\;\left(n\right)=\sum_{n=0}^{N}{\mathrm{FSR}}_{\mathrm{toe}}\left(n\right)+{\mathrm{FSR}}_{\mathrm{MTH}}\left(n\right)+{\mathrm{FSR}}_{\mathrm{heel}}\left(n\right)$$

To remove baseline offsets within the pressure signal (e.g., due to the lacing of the shoes), the minimum pressure was subtracted from between the manually annotated stride borders. Finally, a threshold-based approach similar to [35] was applied to the total weight signal in order to define the time points of the reference IC and TC. For our work, an individual threshold of 7.5% of the participant’s body weight was chosen empirically, as shown in Figure 6.

### 2.5. Multiclass Hidden Markov Model

#### 2.5.1. Model Architecture

To detect level walking and stair strides and to extract spatio-temporal parameters, respective stride borders must be segmented from the continuous IMU data streams. For this work, a previously published HMM based approach [18], which was already evaluated on continuous real-world walking data, was extended for the purpose of stair-stride segmentation. Therefore, the previous HMM architecture and model parameters were extended to a multi-stride class segmentation model. Since stair strides follow a strict and repeating biomechanical pattern (repeating swing and stance phase) similar to level walking strides, the same left-to-right Markov chain structure could be used to model stair ascending and descending strides. Hence, the respective stride sub-models allow only self-transitions $p_{n,n}$ and transitions to the next hidden state $p_{n,n+1}$. Similar to the previously published HMM, a transition model was defined, which allowed non-walking activities to be included in the final segmentation model. For the transition model, additional transitions within the model were allowed as transitions might not follow a strict repetitive pattern. The resulting transition matrices for strides $W_{i}\in R^{N_{i}\times N_{i}}\;\forall \;i\in \left\{L,A,D\right\}$ with L = **L**evel walking, A = **A**scending, D = **D**escending, and $T\in R^{N_{T}\times N_{T}}$ for **T**ransitions were defined as follows: (6)$$\begin{array}{c} W_{i}=\left[\begin{array}{ccccc} p_{0,0}^{i} & p_{0,1}^{i} & 0 & \cdots & 0\\ 0 & p_{1,1}^{i} & p_{1,2}^{i} & \ddots & \vdots \\ \vdots & \ddots & \ddots & \ddots & 0\\ \vdots & & \ddots & p_{n-1,n-1}^{i} & p_{n-1,n}^{i}\\ 0 & \cdots & \cdots & 0 & p_{n,n}^{i} \end{array}\right],\\ \;T=\left[\begin{array}{ccccc} p_{0,0}^{t} & p_{0,1}^{t} & \cdots & p_{0,n-1}^{t} & p_{0,n}^{t}\\ p_{1,0}^{t} & p_{1,1}^{t} & p_{1,2}^{t} & 0 & 0\\ \vdots & 0 & \ddots & \ddots & 0\\ p_{n-1,0}^{t} & \vdots & \ddots & p_{n-1,n-1}^{t} & p_{n-1,n}^{t}\\ p_{n,0}^{t} & 0 & \cdots & 0 & p_{n,n}^{t} \end{array}\right] \end{array}$$

Together with the transition model, the final multiclass segmentation HMM was built from four individual trained sub-models: *transition model*, *level walking stride model*, *stair ascending stride model*, and *stair descending stride model*, as illustrated in Figure 7.

To combine the three individual stride models and the transition model in a connected segmentation model, the respective transition matrices were joined and missing edges connecting the sub-models were added. These missing edges (Table 2) correspond to the annotated stride borders and include, for example, self-transitions within a stride model for repeating strides of the same type or between a stride model and the transition model before gait initiation or gait termination. The final **S**egmentation model transition matrix $S\in R^{N_{S}\times N_{S}}$ had a dimension of $N_{S}=\left(N_{T}+N_{L}+N_{A}+N_{D}\right)$:(7)$$S=\left[\begin{array}{cccc} T & 0 & 0 & 0\\ 0 & W_{L} & 0 & 0\\ 0 & 0 & W_{A} & 0\\ 0 & 0 & 0 & W_{D} \end{array}\right]$$

#### 2.5.2. Model Training

Model training was performed as a two-step process: first, each sub-model was trained individually; second, the trained sub-models were combined to create the final segmentation model by adding missing edges, as described in the previous section. As an input for the HMMs, the respective sensor data was first transformed into a feature space. First, the angular velocity in the mediolateral direction ($gyr_{ml}$) was considered since this axis describes the characteristic flexion of the foot during level walking and stair ambulation. Additionally, the acceleration in the superior-to-inferior direction ($acc_{si}$) was added to better distinguish different stride types, with the *si*-axis being aligned to gravity during the sensor-to-body-frame transformation. Other tunable hyperparameters were directly derived from previous work [18], where feature and model parameters were thoroughly grid-searched within a cross-validation. Hence, for the feature space transformation, the data of the $gyr_{ml}$ and $acc_{si}$ axes were downsampled to 51.2 Hz (factor 4) and low-pass filtered with a 4th order Butterworth filter with a cut-off frequency of 10 Hz. As features, the raw data itself as well as a sliding centred-window linear gradient with a window size of 200 ms were calculated per axis. The resulting 4D feature input was finally z-score standardized per axis of each walking bout/task within the dataset. Each state of the HMM was modelled by an individual Gaussian Mixture Model with eight components. The number of states were chosen to $N_{T}=5$ and $N_{L}=N_{A}=N_{D}=20$. Each sub-model was solely trained on the data of its respective class and annotated reference stride borders using ten iterations of the Baum–Welch algorithm.

#### 2.5.3. Stride Border Prediction

To segment strides from unseen IMU data, the hidden state sequence had to be predicted based on the combined and finalized multiclass segmentation HMM using the Viterbi algorithm. From the predicted hidden-state sequence, stride borders were extracted (Figure 8), which correspond to the previously defined state transitions between the sub-models (Table 2), as mentioned before.

To transform the predicted stride borders from the downsampled feature space back into the original IMU data, the respective borders were multiplied with the downsampling factor during the feature transformation. To refine the stride borders, each border was set to the respective minimum in the original $gyr_{ml}$ axis in a window of ± 150 ms around each detected stride border.

After stride-border prediction using the multiclass HMM, the same post-processing rules as those for the manual stride annotations were applied to eliminate explicit false-positive stride candidates (min TC dip and min swing peak; refer to Section 2.4.1). Additionally, a valid stride must have a duration between a minimum of 0.4 s and a maximum of 2.5 s. This range includes the shortest (during fast speeds) and the longest stride times (during slow speeds) within the evaluation dataset, as extracted from the reference labels.

Although the multiclass HMM could already provide stride class labels (*level*, *ascending*, and *descending*) according to the respective hidden-state sequence and sub-models (Figure 8), usage of those labels was omitted, and only the stride borders without any class information were used as an input for the subsequent pipeline steps. The HMM-based classification failed in particular for staircase C, and reached an overall accuracy of 91.2% on the evaluation dataset. In order to maintain readability and the overall structure of the paper, this manuscript does not provide a detailed assessment of HMM-based classification, but only evaluates its segmentation performance. The actual classification task was finally performed based on spatial trajectory features, which is described in Section 2.7.

### 2.6. Event Detection and Parameter Extraction

After the segmentation of individual stride borders, respective gait events such as initial contact (IC) and terminal contact (TC) were extracted by detecting unique signal features within the foot-worn IMU signals between the segmentation borders. Therefore, a strict sequence of events for each stride was assumed. The swing phase starts with the TC (or foot lift), followed by a swing maximum and the maximum forward acceleration, and ends with the IC (ground contact). The subsequent stance phase starts with the IC, followed by the mid-stance (MS) event (Section 2.6.2), and ends with the next TC when the cycle repeats with the next stride. Gait parameters such as the swing, stance, and stride times were directly calculated from those events with $t(IC_{i})$ and $t(TC_{i})$ corresponding to the point in time of the respective *i*-th gait event:(8)$$\begin{array}{cc} \mathrm{Stride}\;\mathrm{time}(i) & =t\left({\mathrm{IC}}_{i}\right)-t\left({\mathrm{IC}}_{i-1}\right) \end{array}$$(9)$$\begin{array}{cc} \mathrm{Swing}\;\mathrm{time}(i) & =t\left({\mathrm{IC}}_{i}\right)-t\left({\mathrm{TC}}_{i}\right) \end{array}$$(10)$$\begin{array}{cc} \mathrm{Stance}\;\mathrm{time}(i) & =t\left({\mathrm{TC}}_{i}\right)-t\left({\mathrm{IC}}_{i-1}\right) \end{array}$$

Although the sequence of events is the same for level walking and stair ambulation, event-detection algorithms—which were initially developed only for level-walking gait similar to the algorithms presented by Rampp et al. [37]—could not be directly applied to stair strides as those expect a present heel strike during IC, which is often missing during stair climbing. Therefore, existing event detection methods were adapted for the purpose of robust gait event detection, which also works during the absence of a heel strike and a first ground contact with toes or a flat foot.

#### 2.6.1. Terminal Contact (TC)

TC events correspond to the minimum peak within the mediolateral angular velocity ($gyr_{ml}$) before the prominent swing peak. This peak is basically the location of the predicted stride borders of the multiclass HMM. However, as the respective minimum might be superimposed by high-frequency noise, the TC label was refined to the respective minimum within a low-pass filtered representative of the $gyr_{ml}$ signal in a ± 150 ms window around the initial HMM stride border. For filtering, a forward-backwards Butterworth low-pass filter with a cut-off frequency of 10 Hz (order = 5) was used.

#### 2.6.2. Mid Stance (MS)

As MS is usually a region rather than a specific event within the gait cycle, the MS event was defined as the centre of the window with the lowest total energy within the 3D gyroscope signal [37]. Therefore, a sample-by-sample, sliding-window, 3D angular velocity energy-detector approach (sliding window size = 200 ms) between the max of the raw $gyr_{ml}$ signal and the next TC was applied. The centre of the window with the lowest signal energy within the search region was defined as MS.

#### 2.6.3. Swing and Forward Acceleration Maximum

To define the swing and forward acceleration maximum, a low-pass filtered representative of the mediolateral angular velocity $gyr_{ml,low}$ and posterior-anterior acceleration $acc_{pa,low}$ were considered. Therefore, a Butterworth filter $f_{c}$ = 5 Hz, order = 5, was applied to constrain the signal information to the locomotion band of gait [38]. Within the low-pass filtered signals, the swing maximum corresponded to the first prominent peak in the $gyr_{ml,low}$ signal between two consecutive TCs. The maximum forward acceleration was then defined as the maximum within the $acc_{pa,low}$ signal, after the swing maximum and before the MS event per stride.

#### 2.6.4. Initial Contact (IC)

For the IC detection, a search region (highlighted region in Figure 9) was defined as the first 60% of the window, between the maximum forward acceleration and the MS event. The IC was then defined as the maximum peak of the squared $acc_{pa}$ signal to enable a sign-invariant detection of the IC.

Since the impact peak during the IC within the $acc_{pa}$ was not always present for every stride (see the last stride in the ascending example in Figure 9), a fallback condition based on the maximum of the derivative of the $acc_{pa,low}$ signal within the IC search window was used. As a condition for a valid peak, a minimum amplitude of 4 g${}^{2}$ was defined to ensure a clear differentiation between the impact and forward acceleration peak.

### 2.7. Stride-Type Classification

A simple but robust approach to differentiate between level walking, stair ascending, and descending constitutes the use of the trajectory features of a respective stride, as presented by Song et al. [28]. During stair ambulation, a stride should end with a positive or negative change in height for ascending or descending, respectively. During level walking, the change in height is expected to be roughly equal to zero between ground contact phases. Therefore, the change in the height of the stride is a prominent feature in the classification of stair strides. Furthermore, the travelled trajectory’s inclination is considered a second feature. The inclination incorporates the stride length, which is usually shorter for stair strides compared to level walking due to the constraints given by the geometry of the stairs, as illustrated for an ascending sequence in Figure 10.

### 2.8. Walking Bout Trajectory Reconstruction

To extract the desired spatial stride parameters from the foot-worn IMU data, a strap-down double integration of the 3D acceleration after the compensation of the gravitational acceleration is usually applied. In this case, the gyroscope data serve primarily for orientation estimation [28,37,39]. However, due to the imperfections of low-cost IMU sensors such as noise, bias, scaling error, sampling imperfections, and others, errors accumulate rapidly during the double-integration process [39]. To counteract these errors, so-called zero velocity updates (ZUPTs) are usually utilized where the foot has near-zero velocity during ground contact. During the respective ZUPT periods, the accumulated drift in velocity can be effectively compensated.

#### 2.8.1. Zero Velocity Update

For the presented pipeline, ZUPT updates were implemented as a combination of a literature-based sliding window acceleration magnitude (${\mathrm{ZUPT}}_{\mathrm{acc}}$) and a gyroscope energy detector (${\mathrm{ZUPT}}_{\mathrm{gyr}}$), with a window size of 150 ms [40]. Threshold values of 10 dps for the ${\mathrm{ZUPT}}_{\mathrm{gyr}}$ and 0.1 g for the ${\mathrm{ZUPT}}_{\mathrm{acc}}$ detector were chosen empirically for the evaluation dataset.

Based on the segmented stride borders and extracted gait events, an additional ZUPT condition was introduced, corresponding to a 50 ms window around the detected MS event (${\mathrm{ZUPT}}_{\mathrm{MS}-\mathrm{event}}$). Therefore, a minimum ZUPT period was enforced for each segmented stride even if the literature-based ZUPT detectors did not indicate a respective ZUPT window. This was especially the case during fast stair ambulation, where only minimal ground contact phases were present. Finally, all ZUPT phases (boolean arrays) were simply added together (=logic OR operation).
(11)$$\mathrm{ZUPT}={\mathrm{ZUPT}}_{\mathrm{acc}}+{\mathrm{ZUPT}}_{\mathrm{gyr}}+{\mathrm{ZUPT}}_{\mathrm{MS}-\mathrm{event}}$$

#### 2.8.2. Error-Tracking Kalman Filter

The sensor/foot trajectory (Figure 11) was then reconstructed using an error-tracking Kalman Filter (ETKF). The ETKF implementation was based on the work of Tunca et al. [39]. After the initialization of the filter using an accelerometer-based orientation update during a static period, the 3D trajectory was calculated by applying the inertial strapdown navigation equations. During this part, the Kalman filter tracked the nominal state (position, velocity, and orientation) without any corrections and only updated the respective error-state covariance at each time step based on the process noise. When complementary measurements were available during the ZUPT phase, the difference between the nominal state and the measurement was calculated and the error-state was updated accordingly. Note that the nominal state was not corrected using the updated error state; instead, the filter was run open-loop [41]. To counteract any trajectory discontinuities during the ZUPT update, a Rauch–Tung–Striebel (RTS) smoothing was used to adjust the error state and the respective covariance. The final "smoothed" error state was then used to correct the nominal state and derive the final estimates for the position, velocity, and orientation of the sensor. The level walking/zero-z update of the original work by Tunca et al. was not included in our approach as a zero-z assumption would obviously be violated during stair ambulation. Because the z-direction is independent of the sensor heading and only relative stride length values were required, no heading correction was needed for our approach.

Although the ETKF-based trajectory reconstruction did not require a valid orientation update during each MS event, some initialization updates were still required. Therefore, a walking bout was split into smaller subsequences where a reliable orientation update was possible based on static accelerometer readings. To ensure the validity of respective orientation updates, windows with a minimum of 300 ms were considered where the accelerometer variance was below a predefined threshold (0.015 g${}^{2}$) for each individual axis. The orientation was then derived from a low-pass filtered acceleration signal at the centre of the respective window. To ensure a stable trajectory before the first and after the last stride, an orientation update was enforced at the beginning and end of each walking sequence. Finally, all trajectory sub-sequences were added together to form the final walking bout trajectory (Figure 11).

#### 2.8.3. Spatial Stride Features

Spatial stride parameters were derived directly from the change within the reconstructed trajectory between two consecutive MS events. During MS, the foot’s trajectory can be assumed to be static for a short period of time, with $S_{d,i}\;\forall \;d\in \left\{x,\;y,\;z\right\}$ corresponding to the respective 2D projection of each world frame direction (*x*-, *y*-, and *z*-axes) for the *i*-th stride: (12)$$S_{d,\;i}={\mathrm{Trajectory}}_{d}\left({\mathrm{MS}}_{i}\right)-{\mathrm{Trajectory}}_{d}\left({\mathrm{MS}}_{i-1}\right)\;\forall \;d\in \left\{x,\;y,\;z\right\}$$

Based on the individual world frame trajectory parameters, stride parameters can be reconstructed, with i corresponding to the respective stride index.
(13)$$\begin{array}{cc} \mathrm{Stride}\;{\mathrm{height}}_{i} & =S_{z,i} \end{array}$$
(14)$$\begin{array}{cc} \mathrm{Stride}\;{\mathrm{length}}_{i} & =\sqrt{{S_{x,i}}^{2}+{S_{y,i}}^{2}} \end{array}$$
(15)$$\begin{array}{cc} \mathrm{Stride}\;{\mathrm{inclination}}_{i} & =\arctan\left(\frac{\mathrm{Stride}\;{\mathrm{height}}_{i}}{\mathrm{Stride}\;{\mathrm{length}}_{i}}\right) \end{array}$$

### 2.9. Walking Bout Assembly

As a final step of the proposed analysis pipeline, the parametrized and classified strides were combined into walking bouts. From such walking bouts, final DMOs such as average gait characteristics or variability parameters can be extracted to quantify real-world walking behaviour [42]. Kluge et al. [42] defined a valid walking bout as a sequence of at least two consecutive strides of the left and right feet, respectively (e.g., L-R-L-R or R-L-R-L, with L/R being parametrized strides from the left (L) and right (R) feet). We extended the proposed walking bout definition from Kluge et al. for stair ambulation, adding the rule that for a valid stair walking bout, a minimum number of consecutive strides must have the same type (A = **A**scending, D = **D**escending, and LW = **L**evel **W**alking). Therefore, given a stride sequence of L_A_-R_A_-L_A_-R_LW_-L_LW_-R_A_-L_A_, only the ascending strides R_A_/L_A_ fulfill the requirements to form a valid walking bout and the intermediate level walking strides L_LW_/R_LW_ are not included within the final DMOs. In the opposite case *…*-R_LW_-L_LW_-R_D_-L_D_-R_LW_-L_LW_-*…*, which might occur when making a single-step transition such as at a curbstone, or when ascending/descending a two-step staircase at a house entrance, these strides do not form a valid stair-walking bout and will therefore be treated as level-walking strides. Thus, a high precision for the classification of stair strides can be guaranteed, as those need to fulfil the individual requirements for stride height and inclination, and additionally need to be included within a valid stair bout.

## 3. Evaluation

### 3.1. Stride Segmentation

For the evaluation of the HMM stride segmentation performance, a leave-one-participant-out cross-validation on the evaluation dataset was performed to avoid any train-test leakage. Hence, a new model was trained for each participant, considering all annotated strides (all 21 tasks in all speeds) from the remaining 19 participants. Using this model, stride borders were predicted on the unseen IMU data of the respective participant under evaluation. The segmentation performance was evaluated in terms of precision, recall, and the F1-score. A stride was considered true positive (TP) if the segmented start and end borders of the stride were within a window of ± 100 ms compared to the ground truth borders. If one of the two borders were not within this margin, the stride was considered a false positive (FP). Each stride from the manually annotated ground truth labels, where no matching stride was predicted by the HMM, was considered a false negative (FN). The final scores were then calculated for each participant of the evaluation dataset as follows:(16)$$\begin{array}{c} Precision=\frac{TP}{TP+FP},\;Recall=\frac{TP}{TP+FN} \end{array}$$(17)$$\begin{array}{c} F1-score=2\;\cdot \;\frac{precision\;\cdot \;recall}{precision+recall} \end{array}$$

### 3.2. Event Detection

To evaluate the accuracy of the proposed event detection, the ground truth stride borders together with the raw IMU data were considered an input to ensure that for each stride, a valid reference existed since for a potential FP-segmented stride of the HMM output, no timing error for respective events can be calculated. As an evaluation criterion, the timing error of the predicted IC and TC events from the IMU data as compared to the pressure insole events was assessed. Additionally, respective temporal gait parameters (swing, stance, and stride times) were evaluated, which were derived from the respective events. An additional post-processing step was added to exclude potential technical outliers. Such outliers were defined as strides which resulted in biomechanically unrealistic values and should therefore be excluded from any final parameter assessment. A stride was considered valid if the predicted absolute swing and stance times (T_swing_, T_stance_) as well as the relative swing time proportion (R_swing_ = T_stride_/T_swing_) were within biomechanically reasonable ranges: 0.2 s≤ T_swing_≥ 1.0 s, 0.2 s≤ T_stance_≤ 1.5 s and 25% ≤ R_swing_≤ 60%.

### 3.3. Stride-Type Classification

The thresholds for the stride-type classification were chosen according to the stair step geometries (refer to Figure 1) to include single and double stair step strides. Based on those requirements, the thresholds were a minimum stride height of ± 10 and a minimum inclination of ± 6 ${}^{\circ }$ for ascending and descending strides, respectively.

Likewise, to evaluate the ability of the trajectory-based approach in rejecting potential FP strides due to biomechanically unreasonable trajectory values (e.g., a stride length of less than 25 cm or more than 200 cm), the predicted stride borders of the HMM segmentation block and respective MS events were considered for the evaluation. As no class reference for the FP or FN segmented strides was available, those strides were assigned to a respective null class.

After the classification, the previously mentioned walking bout assembly was also applied, which ensures a high precision for the detection of stair strides as a minimum of five consecutive ascending or descending strides (e.g., L_A_-R_A_-L_A_-R_A_-L_A_) need to be classified correctly to form a valid stair bout. In order to be considered consecutive strides, stair strides must not extend a distance of more than one maximum stride time ( 2.5 s). This condition should avoid splitting stair sequences due to single-level walking strides, which might appear, for example, during short plateaus (like present in staircase A and B) or single misclassifications.

### 3.4. Full Pipeline Validation

To fully validate the proposed pipeline, respective DMOs, which are of interest for clinical analysis, were assessed. Therefore, all of the aforementioned and individually evaluated steps of the pipeline were combined, including segmentation, event detection, trajectory reconstruction, trajectory feature-based classification, and walking bout assembly. A full evaluation of the pipeline is necessary as respective pipeline steps build upon each other. Therefore, errors might be propagated throughout the whole pipeline, for example, when a falsely segmented stride candidate gets parametrized and included in the final DMO. On the other hand, an FP stride might also be excluded during parametrization, resulting in biomechanically unrealistic parameters, which were derived by the subsequent steps of the pipeline.

Therefore, the pipeline was validated end-to-end, taking raw IMU sequences as input, (e.g., from future real-world gait analysis studies) and providing clinically relevant DMOs as an output. The entire, pipeline including individual processing blocks, is illustrated in Figure 12. In this work, respective DMOs were the mean and standard deviation of temporal stride parameters (stride, swing, and stance times) per individual gait activity (stair ascending, stair descending, and level walking).

For the final validation, the combined task for each of the three staircases in the evaluation dataset was considered. The task was chosen because it resembled the most realistic real-world scenario where participants went up and down the staircases within continuous walking sequences and at their preferred walking speed. Therefore, all three gait activities were present in one continuous recording.

Additionally, the pipeline was validated on the PD patient validation dataset to test the validity of the proposed pipeline on the data of elderly patients with potential gait impairments. For this dataset, one combined task per participant was available, which resembled one transition between floors within the hospital.

## 4. Results

### 4.1. Stride Segmentation

Segmentation results were assessed per task within the evaluation dataset. Therefore, each task (ascending or descending) contained some additional level walking strides due to the transition phase before and after the stair walking, or even all three stride types for the combined task, which included level walking, ascending, and descending within one continuous walking sequence. Tasks were grouped according to walking speed and stair direction, with every group containing the results of all three recorded stair configurations. The results are summarized in Figure 13.

Overall, the proposed HMM achieved an average segmentation F1 score of 98.5% for the complete dataset and all tasks. The ascending and descending tasks showed slightly better segmentation results during the preferred gait speed as compared to the slow and fast speeds. While the performance in average segmentation results between gait speeds was comparable for the descending tasks, reaching values between 98.2% (fast) and 99.0% (preferred), some differences related to speed were found during stair ascending. Here, the worst performance was found in the slow ascending task, with a mean F1 score of 95.1% and a 4.9% interquartile range (IRQ), compared to 98.6% with 1.8% IRQ for preferred speed. These differences were mainly related to three individual participants (hence, the increased IRQ of 4.9%), where the TC peak was no longer clearly defined during slow ascending, showing two neighbouring minima with a distance of slightly more than the accepted tolerance of ± 100 ms. Here, the HMM sometimes detected the wrong peak compared to the manual labels, and the stride was consequently considered an FP. The performance at the preferred gait speed showed the best results across all tasks as participants presumably walked more naturally and were therefore more consistent compared to their performance at the slow or fast speeds. The adapted speeds (individually interpreted by each participant) also strongly varied between participants, with some walking either extremely slow or reaching almost running speed during the fast task. The leading performance in terms of the average F1 score of 99.6% during the combined task was certainly related to the presence of long, mostly straight level walking sequences between the actual stairs, which the HMM reliably segmented. Overall, the proposed multiclass HMM was able to robustly segment strides within varying gait activities (including transitions) on different stair geometries at varying gait speeds.

### 4.2. Event Detection

The event detection block reached a detection rate of 99.0%. Accordingly, only 1.0% of the strides had to be excluded during the post-processing due to biomechanically unrealistic values, as previously described.

The TC and IC timings, as well as the temporal gait parameters extracted from the IMU data, were in good agreement with the pressure sensor reference (Table 3 and Figure 14). All gait events could be extracted with a mean timing error of below ± 10 ms, a standard deviation of below 29 ms, and mean absolute errors below 20 ms. The largest error of 9.8(269) ms was found for the IC event during stair ascending. However, considering the maximum temporal resolution of the proposed system ( 4.9 ms at 204.8 Hz sampling frequency), this error corresponds to only two samples on average.

The extracted temporal parameters showed a similar error range, as those parameters are directly derived from the respective gait events. The stride time (IC to IC) could be reconstructed with a mean error below ± 0.4 ms for all three stride types. The minor systematic errors for IC and TC timings were also reflected in the swing and stance times, with the worst results for stair ascending showing mean errors of 11.2 ms and −11.0 ms for swing and stance times, respectively, with the highest standard deviation found at approximately 40 ms. All timing errors were in an acceptable range, close to the overall temporal resolution of the system.

First, a good agreement between the extracted parameters and reference values was found for all parameters and gait activities (Figure 14). Second, the direct correlation between temporal parameters and stair-walking velocity is clearly visible for ascending and descending strides. This is related to the fact that a given stair geometry constrains specific spatial parameters such as stride length and stride height. Hence, the gait velocity on the stairs is mainly modulated by extended or shortened swing and stance times, while for level walking, the stride length is also adapted in addition to the stride time. Therefore, differences in temporal parameters between speeds are not as pronounced for level walking as compared to stair ambulation.

### 4.3. Spatial Stride Features

The extracted spatial parameters per stride show a good separability between individual stride types, as illustrated in Figure 15. Double stair step strides on staircase A and B form visible clusters according to the specific stair geometries, while double stair step strides of staircase C are scattered across a greater range due to the unique stair geometry of long and flat steps. Single stair step strides show a reduced stride height (as those correspond to only a single step) and a wider spread in length (as for transitions, the stride length is no longer constrained by the fixed stair geometry).

### 4.4. Stride-Type Classification

Figure 16 shows the confusion matrix for the classification of level walking, stair ascending, and stair descending strides for the spatial stride feature threshold-based classification approach. An additional null class contains FP and FN-segmented strides where no reference stride-type label could be assigned. Detailed results in terms of precision, recall, and F1 score are summarized in Table 4. Overall, the classification block using trajectory features (stride elevation and inclination) and the simple threshold-based decision rules achieved a total balanced accuracy of 98.2%.

The combined HMM + trajectory approach reached a precision of 100% for detecting stair strides with only a single misclassification of a level walking stride as an ascending and a descending stride. Furthermore, not a single ascending stride was confused with a descending stride in all three stair configurations and walking speeds, which was additionally ensured by the subsequent walking bout assembly requirements. Although some stair strides were confused with level walking strides, the misclassification rate was below 3% for ascending and descending strides. Misclassifications happened mostly during transition strides as well as on staircase C due to the unique step dimensions, which were closest to the classification thresholds (height = 13 cm and inclination = 7.6 ${}^{\circ }$).

The number of FP segmented strides could be reduced by 26%, from a total of 419 to 310 strides, when adding trajectory features to the HMM output by utilizing basic biomechanical exclusion criteria such as unrealistic spatial or temporal parameters.

### 4.5. Full Pipeline Validation

Overall, the pipeline was in good agreement with the ground truth reference (Figure 17), with a mean difference of parameters for the mean and the standard deviation of walking bouts below 0.01 s. The 95% limits of agreement were below ± 0.05 s for the evaluation dataset and only marginally higher with ± 0.06 s for the PD validation dataset. The results indicate good generalization and robustness of the proposed pipeline, with similar error ranges for both datasets and potential application on the data of patients with gait impairments, as in PD.

## 5. Discussion

Extracting objective and clinically relevant parameters from continuous IMU data is not a trivial task. Several algorithmic steps are necessary and depend on each other to form a complete gait analysis pipeline. The aim of the presented work was to extend state-of-the-art gait analysis systems to real-world assessments where additional gait-related activities such as stair ascending and stair descending are a fundamental part of daily living. As stair ambulation adds unique challenges compared to level-walking gait, respective objective stair ambulation parameters should be included in future clinical analyses, which would then require adapted algorithms for segmentation, parametrization, and classification. To address the lack of stair ambulation analyses in continuous real-world studies, a new analysis pipeline targeting foot-worn IMUs was presented. Each pipeline step was adapted to enable the simultaneous parametrization and classification of gait-related activities, including level walking, stair ascending, and stair descending. Therefore, an extended multiclass HMM for robust level walking and stair-stride segmentation, an adapted event detection block to extract temporal stride parameters independent of the gait activity, and precise spatial parameter extraction for stride-type classification using an ETKF-based approach were presented and individually evaluated. Finally, the full pipeline was validated end-to-end with respect to extracted DMOs on 20 healthy participants and 13 PD patients in order to prove its applicability in continuous real-world applications.

### 5.1. Datasets

Although the dataset was still part of a supervised study to enable the required ground truth data for a technical evaluation of the presented methods, we tried to include a wide variety of strides within the dataset in order to resemble real-world gait data as close as possible. Therefore, we collected data from 20 healthy participants as well as 13 PD patients walking on an outdoor course with three very different stair configurations at different speeds for validation. A total of approximately 16,000 stair strides and 13,000 level walking strides were included during the evaluation (including transition strides between activities), while other studies often only recorded data of a single staircase with a limited number of strides or participants [22,23,24].

Since such out of lab studies cannot make use of external reference measurements such as motion capture or force plates, a mobile reference system consisting of pressure insoles and video reference was used. Although using FSR sensors and thresholding methods might be less precise than external reference systems (e.g., motion capture or force plates), an accurate synchronisation of respective data streams was guaranteed due to the direct integration of the pressure sensors and the IMU sensor. Furthermore, each FSR sensor was individually calibrated to ensure the linearity of the pressure data.

### 5.2. Stride Segmentation

The first contribution of the proposed pipeline was the identification and segmentation of individual strides regardless of the underlying gait activity. Due to the changing foot rolling behaviour during stair ambulation, the segmentation methods developed purely for level walking strides were insufficient if stair sequences were included. Hence, an existing HMM [18] for real-world stride segmentation was extended to a multiclass HMM in order to also model respective stair ascending and descending strides. To enable the simultaneous detection of stride borders of different gait activities, individual sub-HMMs per stride type were trained using the annotated data and subsequently combined with the extended segmentation model. The proposed multiclass HMM achieved a promising segmentation performance, with an F1 score of 98.5(11) % for the evaluation dataset. The worst F1 score of 95.1(50) % was found during slow stair ascending where the characteristic TC peak for some participants was no longer clearly defined, and the evaluation criteria for valid stride borders were not met. Compared to previous studies [17,18,19] (which only included level walking gait during evaluation), our presented multiclass HMM segmentation approach reached slightly improved segmentation performance across multiple gait activities. One limitation of the presented study is certainly the lack of other cyclic activities, which might lead to falsely segmented strides. However, due to the subsequent pipeline steps, such FPs could be effectively rejected due to unrealistic temporal or spatial parameters.

### 5.3. Gait Event Detection

Respective gait events were subsequently extracted based on the IMU data and given stride borders. Although gait event detection using IMU sensors is a widely addressed challenge, many different approaches exist in the literature [22,23,24,37]. However, respective algorithms are either only designed for specific gait activities (e.g., only level walking), expect the presence of a heel strike, or are evaluated only in one stair direction—either ascending or descending—and only on a single fixed stair geometry. Therefore, we adapted the gait event detection block using specific signal features to robustly detect gait events such as TC and IC independent of the gait activity or foot rolling behaviour. This was especially necessary for the stair strides as here, the heel strike is often missing during IC, resulting in toes touching first. IC and TC could be detected with mean errors below ± 10 ms in all conditions and mean absolute errors below 20 ms with standard deviations below ± 30 ms, which is comparable to other state-of-the-art event detection studies [22,23,24,37]. Due to the availability of reference labels only for TP strides, potential errors due to FP-segmented strides could not be considered during the evaluation of the event detection. Therefore, our proposed gait event detection is strongly dependent on the correct segmentation of the HMM.

### 5.4. Spatial Features and Stride-Type Classification

To extract spatial features from the IMU signals, an existing ETKF-based approach [39] was implemented. This approach enables the reconstruction of the trajectory across multiple strides without the necessity of estimating a new initial orientation for every single stride as required by stride-based methods [28,37]. Furthermore, instead of employing linear drift models, the Kalman filter can accurately track the accumulating errors and perform respective correction updates during the ZUPT periods. To ensure at least a minimum ZUPT for each stride, a minimum ZUPT phase for each MS event from the previous event detection block was enforced in addition to state-of-the-art ZUPT detection methods. This step was required especially for fast stair walking where literature-based ZUPT conditions were not satisfied. Due to the limited spatial reference data within the dataset, only the measured stair step height could serve as a reference to tune the respective parameters of the Kalman filter. Although the spatial features showed a good separability between the three stride types (Figure 15), hence proving their validity for the classification task, those spatial parameters will require a separate validation for clinical analysis. Therefore, additional studies should be conducted with a motion capture system to track the reference trajectory in all dimensions and evaluate the accuracy of spatial parameters such as stride length or foot clearance during stair ambulation. In particular, the clearance during stair climbing might be an important spatial feature for gait safety while stair walking as low foot clearance might result in tripping or falling, leading to severe injuries. This feature could be easily extracted from the reconstructed trajectory (Figure 18).

Although the final classification was based on only two features and respective static thresholds, the classification block reached promising results, with F1 scores of more than 98 % for all three stride types and a balanced accuracy of 98.2%. For the presented work, we even included single stair step strides, which usually happen during transitions between level and stair walking or during stair landings, while double stair step strides mostly correspond to steady-state stair walking. The feature-based approach enables a flexible adaption of thresholds to include, for example, only a specific type of stair strides. Especially in real-world studies where participants might face slightly different stair geometries, respective thresholds could be adapted to include only strides of stairs that share, for example, a similar slope in order to enable comparable conditions. The presented approach might also be able to distinguish between single and double stair step strides, which form an individual cluster within the derived feature space (as can be seen in Figure 15), which could enable a separate analysis of respective transition strides before and after stair walking. An extension of the stride-type classification differentiating between single and double stair step strides should be evaluated in future studies. Still, spatial features derived from inertial data are subject to errors due to bad sensor calibration, unreliable zero velocity updates, or aliasing and sampling effects, which will lead to classification errors if static thresholds are applied. Such issues might be avoided with the use of more advanced classification methods such as deep learning-based approaches, as proposed in the field of human activity recognition [43], which enable direct classification based on raw IMU data without the need for expert features.

### 5.5. Full Pipeline Validation

As a final step, the entire pipeline was validated end-to-end with respect to DMOs. This step is often overlooked but very important for future real-world analysis applications as reference labels are often missing; hence, retraining or optimizing parameters is difficult or impossible. Furthermore, single pipeline blocks usually depend on each other, which can lead to the propagation of errors but also to the possibility of being cancelled out by subsequent pipeline blocks. Therefore, only an end-to-end validation of all combined steps can capture the pipeline performance under realistic conditions. Especially for real-world studies, which come more and more into the focus of clinical gait analysis [10], application-driven pipelines that can robustly and automatically handle continuous input sequences without manual intervention are required. Respective output parameters in the form of objective and interpretable parameters could then be used to answer clinically relevant questions such as the assessment of a patient’s fall risk and disease state or progression. Because stair ambulation speed is primarily modulated by varying stance and swing times due to the constrained stair geometry (as illustrated in Figure 14), respective parameters were considered as primary DMOs for the final validation. The pipeline showed a good agreement of temporal stride parameters for both the healthy-participant evaluation dataset and the PD-patient validation dataset, with mean errors below 10 ms for average bout parameters as well as their standard deviation. The overall performance was comparable between both datasets and also proved the validity of the presented pipeline for elderly patients suffering neurological conditions such as PD. Although the presented results were promising, the respective datasets were collected under supervision following a predefined protocol and a limited number of individual patients. Therefore, the proposed pipeline will still need to prove its performance on unsupervised continuous real-world data in future studies, where changing environments, underground, or stair geometries, as well as unexpected movements and activities, might lower the accuracy of the extracted gait parameters. Finally, extracted real-world stair ambulation DMOs must undergo clinical validation and prove their applicability in supporting clinical questions such as the assessment of fall risk, disease state, or therapeutic effects.

## 6. Conclusions

We presented a new gait analysis pipeline designed for simultaneous stride segmentation, parametrization, and classification for foot-worn IMU data. The proposed pipeline extends state-of-the-art gait analysis systems by enabling a separate analysis of stair ascending and descending bouts embedded in continuous real-world gait sequences. Each part of the pipeline was thoroughly evaluated on a new dataset of 20 healthy participants containing roughly 29,000 annotated strides at different speeds on different staircases. Finally, the pipeline was validated with respect to DMOs on an independent dataset of 13 PD patients. Our approach showed good agreement with the reference parameters, reaching an average stride segmentation F1 score of 98.5%, with mean gait event timing errors below ± 10 ms for all conditions. Stride types were classified with an accuracy of 98.2% based on spatial features. The end-to-end validation proved the applicability of our proposed pipeline for future real-world gait analysis studies, where different gait-related activities, including stair ambulation, are expected. Here, our pipeline was able to predict temporal DMOs with a mean difference below 0.01 s and ± 0.06 s for the 95% limits of agreement on unseen PD patient data. Thus, we can conclude that foot-worn inertial sensor-based gait analysis systems can accurately measure stair ambulation parameters from continuous gait sequences; therefore, such parameters should be considered in future real-world studies. Due to the unique challenges that stair ambulation poses to the motor and balance control system, the assessment of stair ambulation DMOs could add additional insights into a patient’s mobility behaviour and potential impairments that might not be present during level walking. Overall, the presented work can provide new insights into real-world gait and mobility performance in order to improve clinically relevant outcomes such as fall risk or the monitoring of disease state and progression.

## Figures and Tables

**Figure 1 sensors-21-06559-f001:**
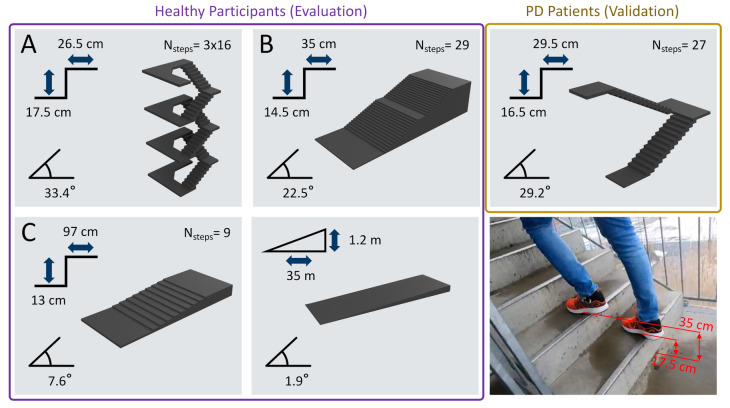
Illustration of the different stair configurations within the presented dataset, including the respective step geometries. (**A**) staircase over four floors including 90 ${}^{\circ }$ turns; (**B**) straight staircase; (**C**) flat staircase with elongated landings.

**Figure 2 sensors-21-06559-f002:**
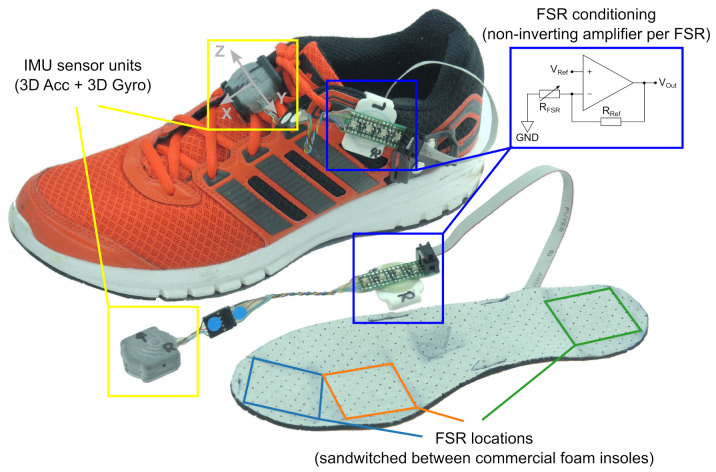
Example of the used sensor setup attached to the left shoe, and the second sensor setup (for the right foot) without the shoe. All participants wore the same Adidas sports shoe model, either in size EU 43 or EU 39.

**Figure 3 sensors-21-06559-f003:**
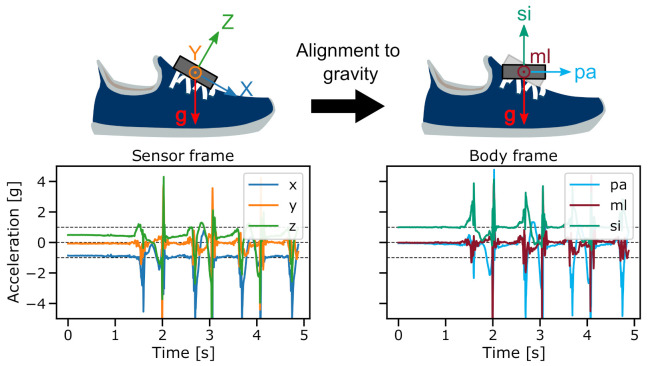
Transformation from sensor frame to body frame by the alignment of the sensor’s *z*-axis to gravity. During static periods, the si-axis of the body frame measures approx. 1 g while the ml- and pa-axes measure approx. 0 g.

**Figure 4 sensors-21-06559-f004:**
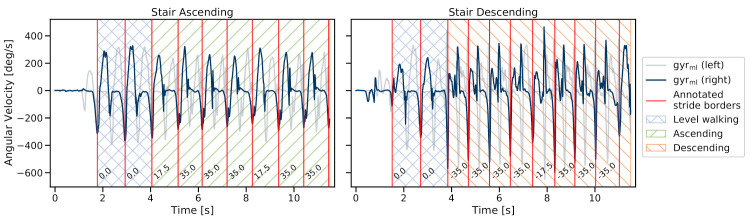
Example of ground truth stride borders and stride class labels. Strides were divided into level-walking, stair-ascending, and stair-descending strides. The numbers for each stride correspond to the respective annotated stair step heights in cm, including single and double stair step strides.

**Figure 5 sensors-21-06559-f005:**
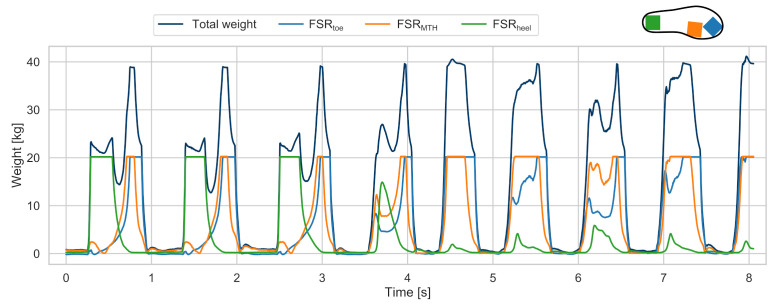
Exemplary FSR data of a gait sequence of level walking with a transition to stair decent. The individual FSR signals from toe, metatarsal head, and heel are combined to a total weight signal for robust event detection. Sensors show a saturation at 20 kg according to the respective calibrated measurement range. During stair descent, a clear forefoot walking behaviour can be seen, with the toes contacting ground first.

**Figure 6 sensors-21-06559-f006:**
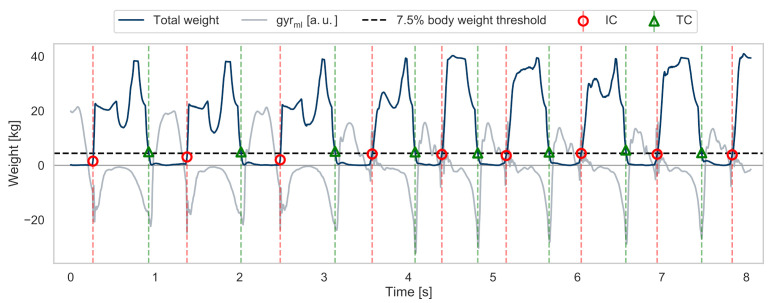
Exemplary gait sequence of level walking with a transition to stair decent. The plot shows the derived total weight signal and the mediolateral angular velocity as reference. Reference gait events were derived by an individual threshold related to the participant’s body weight.

**Figure 7 sensors-21-06559-f007:**
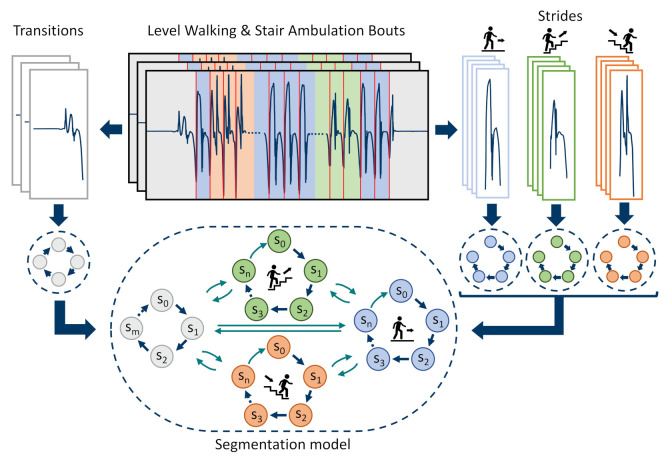
Overview of the multiclass HMM: Three individual stride models (level walking, stair ascending, stair descending), as well as a transition model, are trained based on the annotated walking bouts. Finally, all four models are combined in a final segmentation model by adding missing edges between the sub-models.

**Figure 8 sensors-21-06559-f008:**
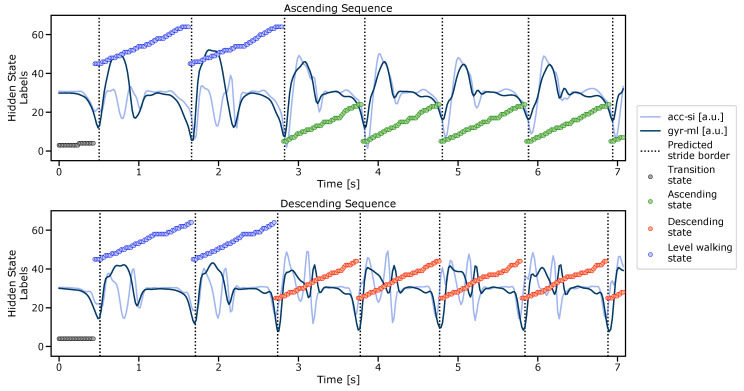
Predicted hidden-state sequence and extracted stride borders using the proposed multiclass HMM. For better visualisation, only the two input streams (raw data, downsampled and low-pass filtered) are displayed.

**Figure 9 sensors-21-06559-f009:**
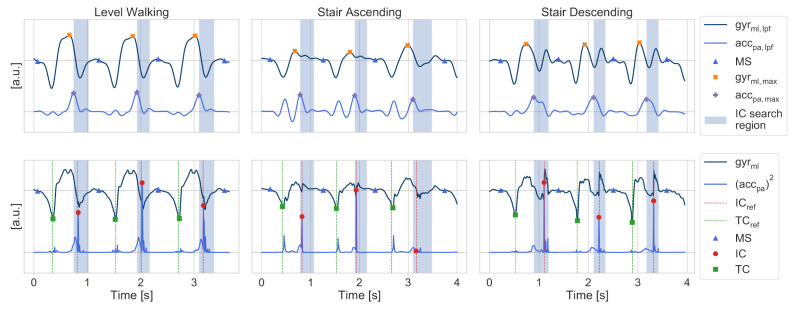
Example strides and corresponding gait-event detection procedure. The first row illustrates the extraction of the swing and forward-acceleration maximum to consecutively refine search regions for the actual gait events. The second row shows the detection of the IC within a previously refined search window.

**Figure 10 sensors-21-06559-f010:**
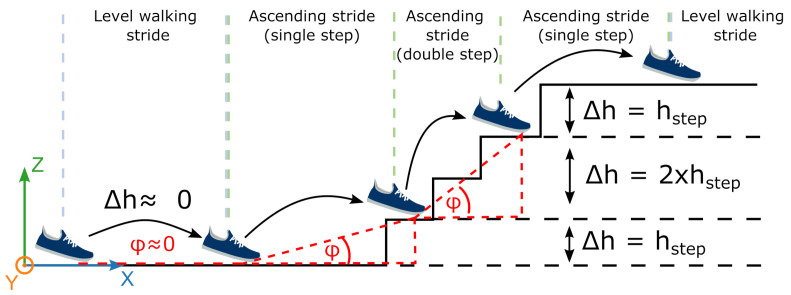
Example of the principle for stride-type classification using trajectory features such as stride height and stride inclination. Both single/transition as well as double stair step strides should be classified as stair strides.

**Figure 11 sensors-21-06559-f011:**
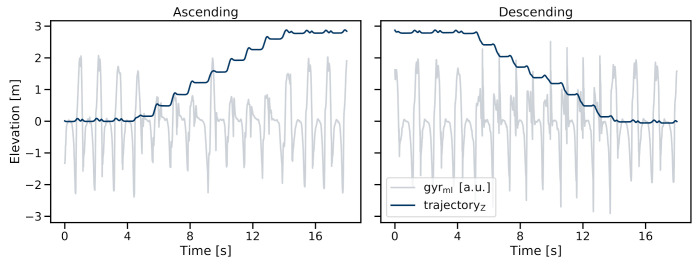
Reconstructed sensor trajectory in the z-direction (parallel to gravity) within the world frame. (**Left**) stair-ascending; (**Right**) stair-descending sequence of one foot-worn IMU sensor. Both stair sequences start and end with level-walking strides.

**Figure 12 sensors-21-06559-f012:**
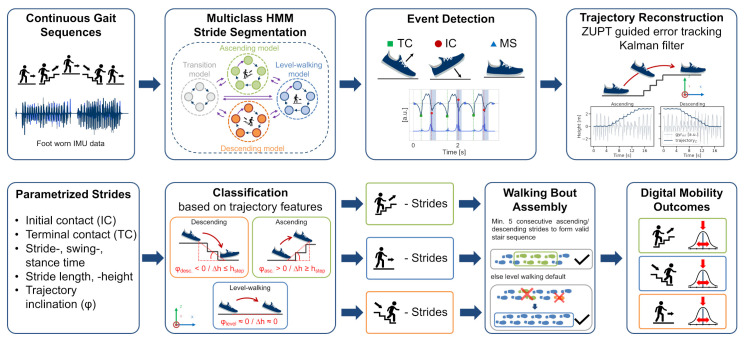
Structure of the final end-to-end validation of the entire proposed pipeline. The pipeline takes raw IMU data sequences as an input (e.g., a real-world walking sequence) and provides DMOs as an output per respective gait activity. A more detailed flow chart is available in the Appendix A.

**Figure 13 sensors-21-06559-f013:**
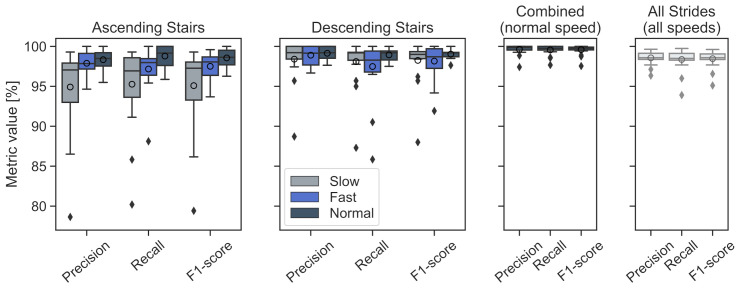
Boxplots of the stride segmentation performance of the multiclass HMM; circles denote the mean performance. Values for every participant of the evaluation dataset (N = 20) are given and grouped according to the respective direction and speed. The combined task includes both stair ascending and descending as well as continuous level walking.

**Figure 14 sensors-21-06559-f014:**
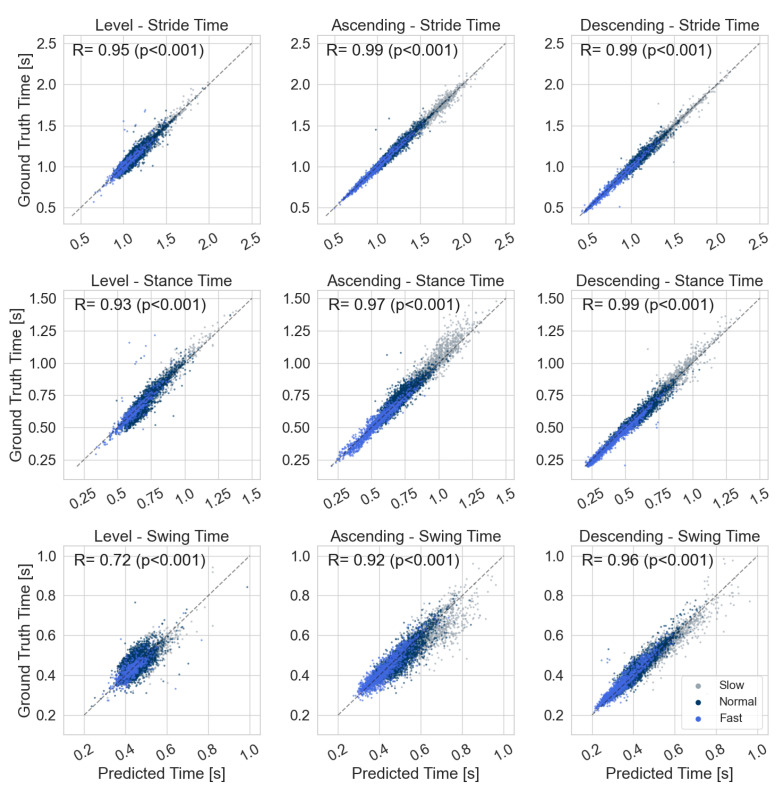
Predicted temporal stride parameters from foot-worn IMUs compared to the ground truth pressure sensor reference. Each scatter plot shows one temporal parameter for one stride type (level walking, ascending, and descending), grouped according to the respective walking speed (slow, normal, and fast). R corresponds to the Spearman correlation coefficient.

**Figure 15 sensors-21-06559-f015:**
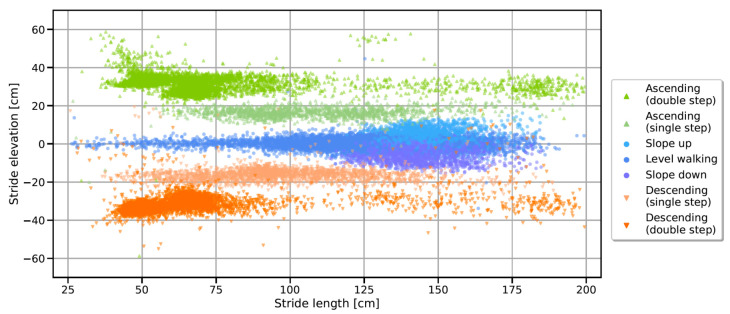
Feature space for stride-type classification. The colours represent ground truth class labels according to the manual video annotations. Visible clusters correspond to the double stair step strides on staircase A (height = 2 × 17.5 cm = 35 cm, length = 2 × 26.5 cm = 53 cm), and staircase B (height = 2 × 14.5 cm = 29 cm, length = 2 × 35 cm = 70 cm). Double stair step strides on staircase C are scattered across the range ≥ 100 cm due to the specific dimensions and comparably long steps.

**Figure 16 sensors-21-06559-f016:**
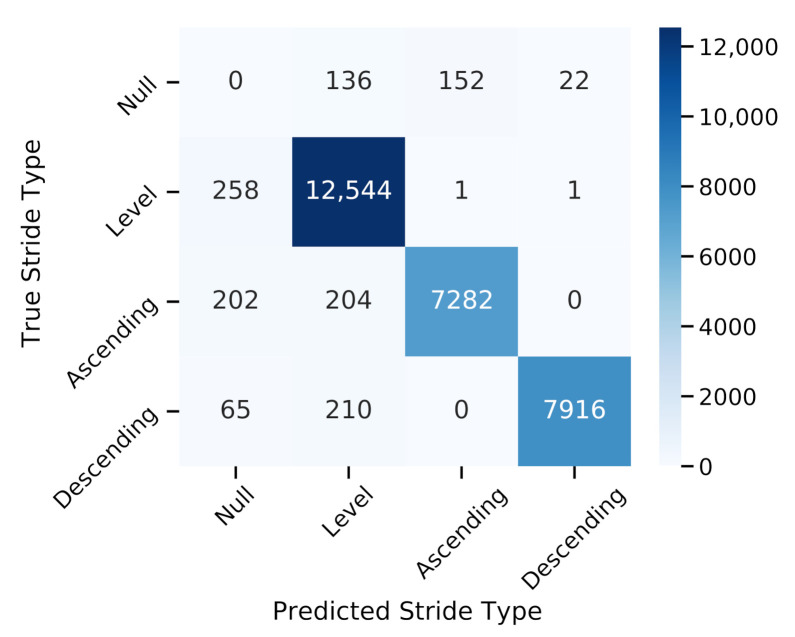
Confusion matrix of the stride-type classification results from the evaluation dataset. Stride types were classified into level walking, stair ascending, and stair descending strides. The null class corresponds to false negative and false positive segmented strides from the HMM-based segmentation output.

**Figure 17 sensors-21-06559-f017:**
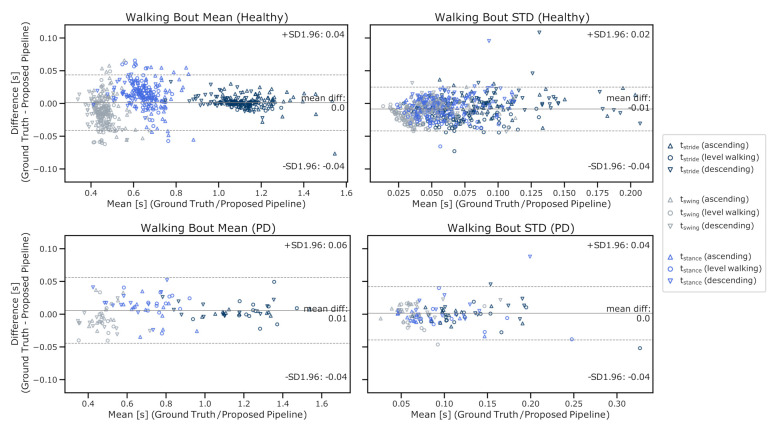
Bland–Altman plots of the full pipeline results in terms of final DMOs. In this case, the mean and standard deviations of respective walking bouts. The *y*-axis shows the difference between the ground truth parameters and the results of the proposed pipeline, while the *x*-axis shows the mean of both methods. Final validation was performed for the healthy (**upper plots**) and the PD datasets (**lower plots**).

**Figure 18 sensors-21-06559-f018:**
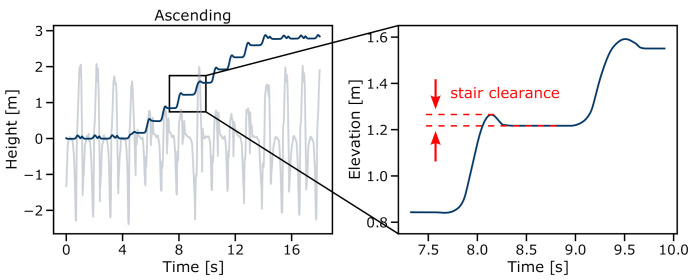
Reconstructed sensor trajectory and respective stair-stride clearance. The distance between the trajectory maximum and the landing flat for each stride could give insights into stair ambulation safety and the danger of tripping or falling.

**Table 1 sensors-21-06559-t001:** Participant characteristics: healthy participants (N = 20) and PD patients (N = 13). Parameters are given by class or by mean ± standard deviation.

Characteristic	Healthy Participants	PD Patients
Gender [f/m]	10/10	3/10
Age [years]	27.1 ± 11.3	61.1 ± 9.0
Height [cm]	173.4 ± 7.1	174.1 ± 8.6
Weight [kg]	68.1 ± 8.6	82.3 ± 15.1
UPDRS-III	-	16.3 ± 7.8
Hoehn & Yahr	-	2.1 ± 0.7

**Table 2 sensors-21-06559-t002:** Possible transitions between sub-models indicate stride borders. $s_{T}$ corresponds to any transition state, while $s_{L_{0}},\;s_{A_{0}},\;s_{D_{0}}$ correspond to the first and $s_{L_{n}},\;s_{A_{n}},\;s_{D_{n}}$ to the last states of the respective level-walking, ascending, and descending stride models.

Stride Type	Label	State Transitions			
Level walking	start	$s_{T}\to s_{L_{0}}$	$s_{A_{n}}\to s_{L_{0}}$	$s_{D_{n}}\to s_{L_{0}}$	$s_{L_{n}}\to s_{L_{0}}$
Level walking	end	$s_{L_{n}}\to s_{T}$	$s_{L_{n}}\to s_{A_{0}}$	$s_{L_{n}}\to s_{D_{0}}$	$s_{L_{n}}\to s_{L_{0}}$
Ascending	start	$s_{T}\to s_{A_{0}}$	$s_{L_{n}}\to s_{A_{0}}$	$s_{A_{n}}\to s_{A_{0}}$	
Ascending	end	$s_{A_{n}}\to s_{T}$	$s_{A_{n}}\to s_{L_{0}}$	$s_{A_{n}}\to s_{A_{0}}$	
Descending	start	$s_{T}\to s_{D_{0}}$	$s_{L_{n}}\to s_{D_{0}}$	$s_{D_{n}}\to s_{D_{0}}$	
Descending	end	$s_{D_{n}}\to s_{T}$	$s_{D_{n}}\to s_{L_{0}}$	$s_{D_{n}}\to s_{D_{0}}$	

**Table 3 sensors-21-06559-t003:** Event detection and timing error characteristics for the evaluation dataset.

Stride Type	Level						Ascending						Descending					
	Mean Error			Mean Abs. Error			Mean Error			Mean Abs. Error			Mean Error			Mean Abs. Error		
IC [ms]	−0.5	±	28.3	15.1	±	26.4	9.8	±	26.9	19.5	±	22.0	1.9	±	23.6	14.9	±	20.5
TC [ms]	−4.0	±	25.9	11.2	±	23.7	−0.8	±	26.2	17.8	±	19.2	6.4	±	14.6	12.0	±	10.4
Swing time [ms]	1.4	±	28.4	19.6	±	20.7	11.2	±	39.2	28.9	±	28.7	−4.5	±	26.3	18.6	±	19.1
Stance time [ms]	−1.8	±	29.5	19.3	±	22.3	−11.0	±	39.6	28.7	±	29.3	4.4	±	26.9	19.1	±	19.5
Stride time [ms]	−0.3	±	31.7	17.5	±	26.4	0.3	±	28.6	18.3	±	22.0	−0.1	±	25.0	14.2	±	20.5

**Table 4 sensors-21-06559-t004:** Stride-type classification results in terms of precision, recall, and F1-score for the classification block using trajectory features (stride height and inclination) and simple threshold-based decision rules.

	Stride-Type Class		
Metric	Level Walking	Ascending	Descending
Precision	96.8%	100.0%	100.0%
Recall	100.0%	97.3%	97.4%
F1 score	98.4%	98.6%	98.7%

## Data Availability

The data presented in this study are available upon reasonable request from N.R. The participants of the presented study did not consent to the publication of their sensor data in open repositories, in accordance with European data protection laws.

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
