# Peer review of "An Inertial Sensor-Based Gait Analysis Pipeline for the Assessment of Real-World Stair Ambulation Parameters"

_sensors, 2021, doi:10.3390/s21196559_

Round 1

Reviewer 1 Report

Line 19~44 This part introduces the behavior of walking too much, can it be simplified?
Line 157~171 Can you describe how sensors including IMU and FSR sensors are installed with an actual picture? And can the circuit diagram of the sensor be disclosed to illustrate how sensors collect datas?
Line 174~187 Process of calculating the angular velocity within the medio-lateral axis with raw IMU data is not mentioned. Is it necessary to explain how to get the angular velocity with formulas?
Line 412~430 As one of the main algorithms, can the process of ETKF be described in more details based on corresponding formulas and specific variables? In addition, why not use other filtering algorithms for trajectory reconstruction?

Reviewer 2 Report

The paper proposes a gait analysis methodology for foot-worn inertial sensors, which can segment, parametrize, and classify strides from continuous gait sequences such as walking, stair ascending, and stair descending. The manuscript should be improved following the comments and suggestions presented below before it could be considered for publication.

  1. What is the knowledge gap bridge by this article? What is your innovation? There are many works published on human activity recognition, including human gait sequence analysis. The novelty to the research area must be explicitly stated in the Introduction section.
  2. The study adopted a Hidden Markov Model (HMM)-based approach Currently, deep learning is used as a modern method for similar knowledge intensive tasks thus avoiding the need for manual construction of features that requires expert knowledge. The deep learning model itself learns the features that are required for classification. Did you consider using a deep convolutional neural network for solving your task?
  3. The discussion on related works should be improved. Introduce a wider context by discussing various problems that require analysis of gait features such as activity recognition, health diagnostics, biometrics. Discuss how similar problems are solved in related domains, see for example doi:10.1016/j.future.2018.02.009, doi:10.1016/j.cviu.2018.01.007 and doi: 3390/diagnostics11081395, for guidance. Discuss the limitations of the existing methods as a motivation for a new method proposed.
  4. Present a description of your methodology as an Algorithm written in pseudocode or as a workflow diagram. Figure 11 illustrates the pipeline, but it is rather informal. A more technical diagram could be added to supplement it and present more technical details.
  5. HMM is an old and well-known method. What is the motivation for selecting HMM as there are many alternatives proposed? Perhaps, you can motivate your choice by supporting references from the HAR domain?
  6. In table 3, confidence intervals exceed 100%. Since the performance data is likely to be not normally distributed, did you use Fisher’s transformation to normalize it?
  7. When presenting the correlation value (R), also present the p-value of the relationship.
  8. Add the critical discussion section. Discuss the limitations of the proposed method and threats-to-validity of the results.
  9. Improve the conclusions by adding the main numerical results from the experimental part to support your claims.
  10. Number your equations.

Round 2

Reviewer 2 Report

I congratulate the authors for the well-executed revision and recommend the paper be accepted.